# Transcriptional and Histone Acetylation Changes Associated with CRE Elements Expose Key Factors Governing the Regulatory Circuit in the Early Stage of Huntington’s Disease Models

**DOI:** 10.3390/ijms241310848

**Published:** 2023-06-29

**Authors:** Sandra Arancibia-Opazo, J. Sebastián Contreras-Riquelme, Mario Sánchez, Marisol Cisternas-Olmedo, René L. Vidal, Alberto J. M. Martin, Mauricio A. Sáez

**Affiliations:** 1Chromatin, Epigenetic, and Neuroscience Laboratory, Centro de Genómica y Bioinformática, Facultad de Ciencias, Ingeniería y Tecnología, Universidad Mayor, Santiago 8580745, Chile; sandraaran@gmail.com (S.A.-O.); mario.sanchez.rubio@gmail.com (M.S.); 2Programa de Doctorado en Genómica Integrativa, Vicerrectoría de Investigación, Universidad Mayor, Santiago 8580745, Chile; 3Laboratorio de Redes Biológicas, Centro Científico y Tecnológico de Excelencia Ciencia & Vida, Fundación Ciencia & Vida, Universidad San Sebastián, Santiago 8580704, Chile; 4Plant Genome Regulation Lab, Centro de Biotecnología Vegetal, Facultad de Ciencias de la Vida, Universidad Andrés Bello, Santiago 8370186, Chile; contrerasriquelme.sebastian@gmail.com; 5Centro de Biología Integrativa, Facultad de Ciencias, Universidad Mayor, Santiago 8580745, Chile; marisol.cisternas@mayor.cl (M.C.-O.); rene.vidal@umayor.cl (R.L.V.); 6Biomedical Neuroscience Institute, University of Chile, Santiago 8380455, Chile; 7Center for Geroscience, Brain Health, and Metabolism, Santiago 8380453, Chile; 8Escuela de Biotecnología, Facultad de Ciencias, Universidad Mayor, Santiago 8580745, Chile; 9Escuela de Ingeniería, Facultad de Ingeniería, Arquitectura y Diseño, Universidad San Sebastián, Santiago 7500000, Chile; 10Centro de Oncología de Precisión, Facultad de Medicina Universidad Mayor, Santiago 7560908, Chile; 11Laboratorio de Investigación en Salud de Precisión, Departamento de Procesos Diagnósticos y Evaluación, Facultad de Ciencias de la Salud, Universidad Católica de Temuco, Temuco 4813302, Chile

**Keywords:** Huntington’s disease, histone acetylation, polyQ diseases, cAMP response element-binding protein, gene regulatory networks

## Abstract

Huntington’s disease (HD) is a disorder caused by an abnormal expansion of trinucleotide CAG repeats within the huntingtin (Htt) gene. Under normal conditions, the CREB Binding Protein interacts with CREB elements and acetylates Lysine 27 of Histone 3 to direct the expression of several genes. However, mutant Htt causes depletion of CBP, which in turn induces altered histone acetylation patterns and transcriptional deregulation. Here, we have studied a differential expression analysis and H3K27ac variation in 4- and 6-week-old R6/2 mice as a model of juvenile HD. The analysis of differential gene expression and acetylation levels were integrated into Gene Regulatory Networks revealing key regulators involved in the altered transcription cascade. Our results show changes in acetylation and gene expression levels that are related to impaired neuronal development, and key regulators clearly defined in 6-week-old mice are proposed to drive the downstream regulatory cascade in HD. Here, we describe the first approach to determine the relationship among epigenetic changes in the early stages of HD. We determined the existence of changes in pre-symptomatic stages of HD as a starting point for early onset indicators of the progression of this disease.

## 1. Introduction

Within genetic pathologies, there is a group of diseases that involve non-traditional inheritance mechanisms characterized by highly unstable mutations called expansions [1]. Most of these expansions correspond to trinucleotide repeats that include CGG, CAG, CTG, CGG, and GAA, among others. Notably, there are at least 10 neurodegenerative diseases known to be caused by CAG repeat expansions, also referred to as polyglutamine (polyQ) diseases, among which there is Huntington’s disease (HD) [2,3].

HD is an autosomal dominant neurodegenerative disorder caused by the expansion of the PolyQ segment equal to or greater than 36 triplet repeats in exon 1 of the Huntingtin gene (Htt), thereby producing a polyglutamine tract in the Huntingtin protein [4,5,6,7]. HD is characterized by neuronal degeneration and loss of brain striatal tissue [1,6] due to the selective loss of GABAergic projection medium spiny neurons (MSNs), which represent approximately 90–95% of the neurons present in the striatal region [8,9]. The number of CAG repetitions is associated with the disease penetrance and the appearance of the first symptoms, which in most cases, begin between the ages of 30 and 50 years [10,11,12,13]. The expansion of 60 or more CAG segment repeats in the Htt gene results in juvenile HD, which usually develops under the age of 20 [8,11,14,15,16,17]. Characteristic HD symptoms include progressive involuntary movements, and behavioral, cognitive, and neuropsychiatric symptoms, among other manifestations [8,15,18,19].

To date, the precise cellular function of the HTT protein, encoded by the Htt gene, has not been yet elucidated [6,20,21], although its role in intracellular trafficking, membrane recycling, neuronal transport, and postsynaptic signaling has been studied [6,21]. The mutant Huntingtin protein (mHTT) form is associated with increased mitochondrial dysfunction, alterations in energy metabolism, oxidative stress, and abnormal interactions between HTT and other proteins [22], all of which can result in the dysregulation of transcriptional machinery and an altered gene expression pattern. Several key proteins from the transcriptional machinery aggregated in the presence of mHTT are affected in the processes in which they participate, as well as in their function, such proteins include the CREB binding protein (CBP), the specificity protein 1 (SP1), and the TATA-binding protein (TBP), among others, which finally produce an effect on gene transcription [12]. CBP is a ubiquitously expressed protein that possess acetyltransferase activity and acts as a transcriptional coactivator of several transcription factors (TFs), such as the cAMP response element-binding protein (CREB) [23,24,25], binding to genes [19]. It participates in different neural functions such as the stress response [26], as a modulator of synaptic plasticity, and synaptic communication [27], among other processes.

CBP can also activate gene transcription through its intrinsic histone acetyltransferase activity [28], adding an acetyl group to lysine residues (K8) of Histone 4 (H4) or H3K27 (H3K27ac), which regulate DNA accessibility to the transcriptional machinery [28,29,30,31,32]. Studies suggest that abnormal transcriptional regulation plays a role in the development of Huntington’s disease, where CBP sequestration disrupts CRE-mediated transcription [12,33], an aggregation confirmed for HD transgenic model R6/2 [34]. Other studies have shown that decreasing CBP aggregation improves mouse condition in the same model [35].

Huntington’s disease has been extensively studied in symptomatic stages at genomic, epigenomic, and proteomic levels, finding that the lengths of the CAG segment of the Huntingtin gene influence alterations in acetylation levels, which translates into changes in regulatory events and ultimately guides neurodegenerative processes and cellular and glial identity [36,37,38]. At pre-symtomatic stages, it has been mainly focused on physiological effects such as neurogenesis [39], retinal adaptation to light and dark [40], and neuroprotective effects of synaptic modulation [41]. However, there also has been found a relationship between alteration in the gene expression levels and H3K27 acetylation changes in the Elk1 gene promoter in HD [42]. In order to understand how epigenetic modification in pre-symtomatic stages of HD gives shape to the chromatin, using a juvenile HD model, we have studied changes in H3K27 acetylation and their relationship with transcriptional processes in striatal neurons, modelling Gene Regulatory Networks to reveal key factors that govern the transcriptional landscape in the early stages of HD.

## 2. Results

### 2.1. Protein Aggregation and Transcriptional Changes in Huntington’s Disease Are Detectable in Pre-Symptomatic Stages

The alteration of gene expression in striatal neurons is an early event described in HD that starts before neuronal death, being nuclear and cytoplasmatic mHtt inclusion. However, the gene expression profile of the R6/2 HD model before the striatal neuronal loss is not yet described. We first see the presence of mHtt accumulation in striatum tissue at 6, 9, and 13 weeks in the R6/2 model. We observed a progressive increase in mHtt aggregation levels in the striatum and cortex brain region (Figure 1). These results confirm the presence of mHtt before the onset of motor symptoms.

To determine the presence of transcriptomic alterations in the early stages of HD, we performed a differential gene expression analysis. To do so, striatal tissue from both Wild-Type (WT) and R6/2 mice models at 4 and 6 weeks were analyzed (Figure A2, Appendix A). We found 27 differentially expressed (DE) genes in 4-week-old mice when comparing WT mice to the R6/2 mice, with 24 of them downregulated in the HD model, while 1365 DE genes were observed in 6-week-old mice, with 381/984 up and downregulated genes in the mutant condition (Appendix A). Comparing the DE genes of both ages, we found three genes shared (Egr1, Rec8, and Gm12695), downregulated in the HD models (Figure A3). Additionally, a two-factor analysis revealed that 1894 genes change their expression levels, with most of the DE genes found in 4 and 6-week-old mice acting through time and the disease condition (Figure A4 and Appendix A).

For a better understanding of processes in which the DE participate, we performed an enrichment analysis of gene ontology (GO) terms. There was no significant enrichment for DE genes for GO biological processes’ molecular functions at four weeks old. Whereas for six weeks old R6/2 mice, we found enriched GO terms related to cell communication functions that could be associated with neurological events present in the context of HD such as synapse organization, cognition, learning, and locomotory behavior for biological process and channel activity for molecular function (Figure 2A,B and Figure A5). Regarding the two-factor analysis, they present similar functions to those at 6 weeks of age, but we also found unique GO terms related to mRNA and tRNA activity (Figure 2C). For this reason, our next analysis centered only on the comparisons of HD and the control condition for each age.

### 2.2. Decreased Levels of H3K27ac Are Detectable at Early Stages of Huntington’s Disease

As described in other murine models of the disease at symptomatic stages, there exists a variation in the acetylation levels in this disease [36,42]. Thus, in order to determine the epigenetic changes in the early stage of HD, we performed a chromatin immunoprecipitation (ChIP) analysis in striatum tissue samples to determine the level of H3K27ac in the R6/2 mice at 4 and 6 weeks old to determine the impact of CBP sequestration by mHTT. Chromatin immunoprecipitation was validated by qPCR using the GAPDH promoter as a positive control for H3K27 acetylation (Figure A6). When an anti-IgG antibody was used, no acetylation of the corresponding promoter region was detected at both 4 and 6 weeks old. However, when using the anti-H3K27ac antibody, we found variations in the acetylation levels when contrasting the control against the HD model, which was most significant in the 6 week old group. In addition, common H3K27 acetylation peaks were evaluated in control and mutant samples at four and six weeks of age. At 4 weeks old, WT samples showed 129,316 common peaks while the R6/2 samples exhibited 153,605 peaks. Similarly, in the case of the 6 week old WT and R6/2 samples, 126,428 and 146,824 common peaks were observed, respectively (Figure A7 and Figure A8). Subsequently, the presence of H3K27ac on promoter regions was determined using ChIP-seq data from the murine models under study. An amount of 176 differentially acetylated promoter regions were found at 4 weeks old, with 92 and 84 promoters up and down acetylated in the mutant condition, whereas at 6 weeks old, 721 promoter regions were found to be acetylated in this same manner, with 399 and 322 up and down acetylated promoters in the mutant condition (Appendix A), without detecting a metabolic or function enrichment for genes presenting these sites.

When looking for changes in the acetylation levels for putative CREB and CBP binding sites (Appendix A), 635 CRE regions were found to be differentially acetylated at 4 weeks old, with 290 and 345 up and down acetylated sites in the HD model, and 39 of them placed in promoter regions and two in a downstream gene region. On the other hand,1558 differentially acetylated regions were found at 6 weeks old, 654 of them up acetylated and 904 down acetylated in the disease condition, and 750 of them placed in the gene body (including 32 DE genes) and 188 promoters (3 of them differentially acetylated). The enrichment analysis of metabolic pathways and GO terms of genes whose CRE regions are differentially acetylated in both ages identified pathways related to the regulation of biological activity through phosphorylation of proteins (Figure 3A,B and Figure A9) and in the context of neurogenesis and synapses (Figure 3C,F).

### 2.3. Altered Acetylation Levels Reshape the Transcriptional Cascades in Presymptomatic Stages of HD

We next contextualized a general GRN considering the quantification of gene expression carried out in this work for the R6/2 murine model at four and six weeks of age. For the two ages, both conditions are similar according to the number of nodes and edges presented in the networks, with few elements presented in only one condition, most of them in the HD model, being associated with the regulation of transcriptional processes in four-week mice (see Table 1, Figure A10, and Appendix A). Additionally, as expected, some of these changes in the transcriptional network are directed by the CRE-CREB-CBP cascade, which progresses as the age of the mice increases (Figure 4).

Looking for a relationship between differentially acetylated regions and DE genes in 4-week-old mice, we did not find a direct association (Figure A11), but the later mice have TFs with lower acetylation levels in their promoters directing the expression of several downregulated genes in the mutant, including EGR1, PDYN, and other genes that exhibit enrichment of Gene Ontology terms related to dopamine release and regulation of calcium concentration functions (Figure 5).

### 2.4. Master Regulators Guiding the Transcriptional Landscape in HD Presents Changes in Their Expression Profiles

Regarding the determination of Master Regulators (MRs), in the four week old stage, we did not find that TF belongs to the MR family (Figure A12). Moreover, even if STAT6 and EGR1 present a physical interaction, they are not regulated by another MR. For the 6 week networks, several TFs considered MRs due to the dense connectivity between them and physical protein interactions were observed (Figure 6). Within these regulators, 84 are shared for both conditions, while three are defined as MRs only in the HD condition (Trp63, Pparg and Pax6). These three MsR are directing genes presenting altered acetylation and/or expression levels in the mutant mice, mainly associated with lipid biosynthesis processes (Figure 7).

On the other hand, 4 and 24 of these MRs are up/down regulated in the HD condition (Figure 6), which controls the expression of other TFs and genes with changes in their acetylation levels in promoter regions (Figure 8). These target genes present functions related to long-term memory, cell communication, and lipid metabolism.

## 3. Discussion

Huntington’s disease is a neuropathology with no cure and whose treatments focus on improving the patient’s quality of life. Current work related to this disease has mainly focused on late stages using the R6/2 murine model [36,42], while studies on early stages have focused, for example, on neurogenesis [39], retinal adaptation to light and dark [40], and neuroprotective effects of synaptic modulation [41].

In this work, we have addressed how changes in the acetylation patterns in the chromatin shape the transcriptomic landscape in the pre-symptomatic stages of Huntington’s disease that have not been addressed in-depth to date in a genomic/epigenomic context, focusing on the integration of data associated with gene expression and changes in H3K27 acetylation levels. We aimed to increase current knowledge on the early stages of HD, trying to identify those regulatory processes whose functional impairment may serve as an early indicator of the disease.

Our results show that non-symptomatic mice present change in their transcriptomic levels when contrasting the HD model versus the healthy condition. Importantly, this increase in the number of DEG shows a direct relationship, whereas when the mice’s ages increase, the number of DEG increases. Overall, an analysis of the functions and pathways in which these DEGs participate shows an enrichment of functional terms linked to ion transport and learning, terms that are related to the functions performed by the striatum, and which are affected as the disease progresses [43,44]. At four weeks, most of the DE genes are downregulated in the mutant condition, whereas only three genes are upregulated. This indicates that there is yet a small variation in the cellular functions in the early stages of HD. This trend is also confirmed by the lower number of DEG at this earlier time point than at six weeks. When studying the functions carried out by genes overexpressed in HD at six weeks, we observed that several functions implicitly related to neurodegeneration, linked to genes such as Adora2a, Gm12695, and Creb3, are enriched, which somehow is unexpected given that six week old mice do not yet show evident symptoms of HD; however, it has been described that symptoms increase with age [37]. Additionally, a two-factor analysis was performed to determine how age and mutation together affect the transcriptomic changes. DE genes found in this analysis are related to the same neurodegenerative processes, but there are genes defined as DE only in this analysis, whose function is linked to mRNA and tRNA activity, showing impairment in the transcriptional machinery.

Furthermore, it is also striking that three genes, Egr1, Rec8, and Gm12695, are DE in the three analyses; both Egr1 and Rec8 are downregulated in HD, while Gm12695 is overexpressed. Egr1 encodes a TF that participates in transcriptional repression and synaptic plasticity processes [45], which are processes known to be impaired in HD. Rec8 is a protein involved in DNA repair and meiosis, whereas the C1orf87 gene, which is an ortholog of Gm12695, has been linked to abnormal vocalization and Spinal Muscular Atrophyatrophia (https://www.malacards.org/card/spinal_muscular_atrophy_distal_x_linked_3, accessed on 15 June 2023).

Regarding known DEG genes in the HD condition also found in our analysis, Crapser and colleagues found a number of genes at eleven weeks which are downregulated [46]. Among these, the Gad1 and Gad2 genes, which encode enzymes that catalyze the synthesis of GABA, the β4 subunit of the channel MSN voltage-gated sodium Scn4b [47,48] and Slc7a11, which is the astrocyte glutamate transporter, indicate an excitatory/inhibitory balance that is deregulated in this model from early stages of HD [46]. In our analysis, Gad1, Gad2, Scn4b, and Slc7 a11 genes are down-regulated at 6 weeks of age. At acetylation of H3K27 levels, we found differences in the number of peaks when contrasting mutant and control conditions at each age, but it was notably more at 6 weeks. This could be an indicator of the existence of a relationship of changes in the transcriptomic guided by changes in the chromatin landscape. When looking for the closest gene promoters differentially acetylated, we observe that there exists an increase of promoters in these conditions when comparing data at 4 and 6 weeks; however, at each age, the number of promoters up- and down-acetylated are similar. This could give clues about a change in the acetylation patterns driven by CBP due to the interaction with the mutant Huntingtin protein or using other acetyltransferases less favorable for the H3K27, such as P300 [24].

When analyzing these differentially acetylated promoters, we did not find GO term enrichment (molecular function or biological processes), but according to our predictions, at 6 weeks, genes that are being regulated by CRE elements act on neurotransmission pathways, which agrees with previous reports that showed a higher neurotransmitter release in the early stages of the disease. One possible cause for this altered release is the dysfunction and death of MSN striatal neurons, which has been related to the origin of motor disorders associated with this disease [49]. By looking for biological processes in which this subset of CRE-regulated genes is involved, we discovered other functions than those related to neurogenesis. For example, we observed genes such as Slc5a1, Fgfr2, Col25a1, Fgfr2, and Igf involved in the regulation of transporter activity, synapse organization, axonogenesis, GTPase activator, and regulator activity, functions whose malfunctioning can thus be driving the early stages of the disease. Moreover, the alteration in the expression and regulation of these genes can be due to a possible reduction in striatal neurogenesis originated by the cascade effects triggered by the mutant Huntingtin protein, which, in the long run, contributes to the depletion of neurons generated by adults [49,50] and, as a consequence, striatum neurogenesis would be affected [51,52]; also of notable relevance is that these processes are related to synapses, as described previously.

The integration of information about differential acetylation levels in promoters and gene expression levels in the gene regulatory networks shows us that for both time points under study, the transcriptional processes are similar (Table 1). Additionally, although there are no changes in the expression profiles at four weeks for CREB or CBP, but given the regulatory networks we generated, it seems that at both ages, this CRE-CREB-CBP regulatory cascade would be directly guiding the changes in expression patterns. When looking for TF whose genes have reduced acetylation levels and the downregulated genes controlled by them, we were not able to find a direct relationship at four weeks, probably due to the scarce number of altered genes, but this is not the case for 6-week-old mice (Figure 5). The regulatory circuit present only at the six weeks time point we studied suggests a relationship between changes in the chromatin status and the main affected genes in this context. Comparing these results with other studies, we found genes previously related to HD through a differential expression analysis such as Bcl6, Foxp1, and Egr1, that presents changes in their expression profiles, whilst other genes such as Crebbp does not change their expression levels, but present change in their acetylation pattern. These results would indicate that Bcl6 and Egr1 could be key regulators whose expression impairment is behind the disease phenotype in the early stages of HD [53]. Moreover, it is worth noting that these genes also exhibit enrichment of Gene Ontology terms related to dopamine release and the negative regulation of calcium concentration functions. It has been suggested that alterations in dopaminergic transmission can induce choreas in HD, and these TFs with altered acetylation changes in their promoter regions may not only lead to increased dopamine levels within the neuron, resulting in oxidative stress, toxicity, and apoptosis, but also to a reduction in dopamine levels in the pre-synaptic space [54]. This dysregulation of the dopaminergic cascade may be associated with our previous findings of altered genes involved in memory and cognition, as the loss of dopaminergic receptors D1 and D2 are known to contribute to cognitive symptoms in HD [55]. Furthermore, calcium-associated functions have been proposed as major players in HD due to the characteristics of dysregulated levels of this ion and it has also been linked to the dopamine cascade, as previously mentioned [56]. Therefore, in the early stages of HD, classic symptoms of the disease are present at the genomic/epigenomic level, despite the lack of clinical symptoms.

Finally, we could not determine master regulators guiding the changes in striatal tissue neurons at four weeks following our approach. This agrees with our findings regarding differential expression and reduced variation in acetylation in the earliest developmental stage we analyzed. However, we have been able to determine several master regulators at six weeks according to the definition proposed by Davis and Rebay [57], where although they are mostly shared in the control and disease model condition, several of them are encoded by DEG or show changes in their H3K27 acetylation. This is the variation in expression and acetylation on the MR genes that could be ultimately guiding the downstream regulatory cascade, which includes genes that are not DE but present promoters that are differentially acetylated (Figure 6, Figure 7 and Figure 8). Following this line, within this group of MRs, we also found TFs associated with cell cycle control, which could be associated with neuronal death and other processes that occur in the context of this disease. For example, Egr1 [58], Foxo1 [59], Stat1 [60], Eno1 [61], and Creb1 [62,63] are known to be key controllers of cell cycle progression. Among these TFs, the role of Egr1 should be studied in depth due to its differential expression pattern (downregulated in HD model) found in 4- and 6-week old mice and its key role as MRs absent in HD which we have identified in this work. Furthermore, only three regulators are uniquely defined in the HD model: Pparg, Pax6, and Trp63. Pparg and Pax6 have been previously described as involved in HD and neuronal death [64,65], whereas Trp63 is known to be involved in cell cycle control [66] and the differentiation of stem cells [67]. Importantly, deficiency in Pparg has been previously linked to an increase in susceptibility to brain damage [68] and Pax6 is key for the regulation of neurogenesis [69], linking the overexpression of these two MRs in HD to a corrective response to the characteristic neurodegeneration of HD. Since in the early disease stages we have focused on, mice are still mostly asymptomatic, our findings indicate a role for these genes protecting neural damage in early HD. However, genes regulated by these three MR perform functions related to lipid metabolism and biosynthesis, required for cell membrane and myelin membrane growth, and their alterations could be the starting point of neuronal disorders [70].

This work consists of a first approach to determine the relationship among epigenetics changes in the HD and how it could be affecting transcriptional levels occurring in the early stages of Huntington’s disease. Therefore, further work could validate the role of genes coding for the MR we found as to be related to early HD response as shown in the network analysis. Moreover, genes shown to be linked to these MR and their functions would be a starting point to drive the discovery and determination of possible therapeutic targets and treatments to attack this disease, given that current diagnosis relies on symptom detection or genetic tests that are usually performed when there is the genetic background to do so. In addition, it is crucial to establish a connection between alterations in other histone modifications and H3K27ac, taking into account previous studies that have reported changes in the levels of H3K4me3, H3K9ac, H3K12ac, and H3K14ac in HD [71,72,73]. These collective alterations likely contribute to shaping the regulatory landscape associated with the disease. By integrating the modifications in multiple histone marks, we can gain a more comprehensive understanding of the molecular mechanisms underlying Huntington’s disease.

## 4. Materials and Methods

### 4.1. Mouse Strains

Striatal tissue samples were obtained from R6/2 female mice at their fourth and sixth week of life. The R6/2 transgenic model that contains exon 1 of the mHtt gene human was obtained from The Jackson Laboratory with all animal protocols abiding by the animal management protocol approved by the Bioethics and Biosafety Committee of the Universidad Mayor.

### 4.2. Transcriptomic Analysis of Striatal Tissues from R6/2 Mice

Samples of striatal tissue from 4- and 6-week-old R6/2 mice and respective controls were obtained in triplicate. Total RNA was purified by tissue homogenization using Trizol^®^ reagent (Invitrogen, Waltham, MA, USA). The total RNA extraction was digested with DNAse I to avoid contamination of the sample with genomic DNA. The RNA concentration was then measured with the Quant-iT ™ RiboGreen^®^ RNA Assay Kit (Life Technologies, Carlsbad, CA, USA), and its integrity was determined by the Bioanalyzer 2100 kit (Agilent Technologies, Santa Clara, CA, USA). RNA libraries were prepared with the Illumina TruSeq TM RNA Sample Preparation Kit, following the manufacturer’s protocol. The rRNA-depleted RNA was fragmented using divalent cations at elevated temperatures. First-strand cDNA synthesis produced single-strand DNA copies from RNA fragmented by reverse transcription. After second-stranded cDNA synthesis, the double-stranded DNA will be repaired at the 3 ’ends. Finally, PCR ligated the universal adapters to the cDNA fragments to produce the final sequencing library. After library validation was carried out with the DNA 1000 chip (on the Agilent Technologies 2100 bioanalyzer), the samples were pooled in equal concentrations into one pool and run on the Illumina HiSeq equipment for 100 base pair-end sequencing.

### 4.3. Identification of Differentially Expressed Genes in the R6/2 HD Mouse Model

The RNA-seq Raw data identified so far for striatal tissue samples of animal models of HD R6/2 for 4 and 6 weeks was analyzed using a pipeline explained as follows: first, the quality of the RNAseq data readings was analyzed through FastQC https://www.bioinformatics.babraham.ac.uk/projects/fastqc/, accessed on 15 June 2023; Then, adapters were removed with Trimmomatic [74]. After that, we built the genome index for the GRCm38/mm10 mouse reference genome using STAR [75], setting the sjdbOverhang option to 149 (readlength-1). Regarding the read alignment step, it also was performed with STAR, with mapped reads being saved into the BAM format and sorted by coordinates. To perform gene quantification, we used Htseq-count [76] over aligned reads, setting the strand as reverse as contrasted with the infer_experiment script of the RSeqQC package [77] and using the intersection-nonempty mode. Finally, the differential expression of the libraries were analyzed with the Deseq2 tool [78], contrasting mutant against control mice by each time and in a two-factor design (genotype and time).

### 4.4. ChIP Analysis in the R6/2 Model

ChIP-Seq were performed on cross-linked chromatin samples, with the following modifications. The tissues were washed with a cold 1 × PBS buffer, incubated for 10 min with 1% formaldehyde (FA) at room temperature, and washed again with 1 × PBS. For ChIP against chromatin-modifying enzymes, double crosslinking with EGS (ethylene glycol bis) (succinic acid N-hydroxysuccinimide ester; E3257; Sigma-Aldrich, St. Louis, MO, USA) was used; FA crosslinked tissue was incubated with EGS for 1 h at room temperature with gentle agitation, and was washed three times with cold PBS, resuspended in 1 mL of cell lysis buffer (5 mM HEPES, pH 8.0, 85 mM KCl, Triton X-100, and proteinase inhibitors) and homogenized with a Dounce homogenizer (Kimble, Vineland, NJ, USA) (25 times using a mortar). The cell extract was collected by centrifugation at 3000× *g*, resuspended in 0.3 mL of sonication buffer (50 mM HEPES, pH 7.9, 140 mM NaCl, EDTA 1 m, 1% Triton X-100, 0.1% deoxycholate acid, 0.1% SDS and a mixture of proteinase inhibitors), and incubated for 10 min on ice extracts from Chromatin sonicated for 15 min, in cycles of 30 s on and 30 s off in BioruptorPico (Diagenode, Liege, Belgium), centrifuged at 16,000× *g* for 15 min at 4 °C to obtain fragments of 500 bp or smaller. The supernatant was collected, divided into aliquots, frozen in liquid nitrogen, and stored at −80 °C; an aliquot was used for quantification by Qubit fluorometric quantification (Thermo Fischer, Waltham, MA, USA).

Chromatin size was confirmed by an electrophoretic analysis (Figure A1). Extracts of cross-linked chromatin (500 ng) were resuspended in the sonication buffer to a final volume of 500 μL; samples were pre-cleaned by incubating with 2–4 μg of normal immunoglobulin G (IgG) and 50 μL of A/G-agarose protein microspheres (Santa Cruz Biotechnology) for 1 h at 4 °C with agitation for preclearance. The chromatin was centrifuged at 4000× *g* for 5 min, and the supernatant was collected and immunoprecipitated with specific antibodies for 12–16 h at 4 °C. The immune complexes recovered with an additional 50 μL of magnetic beads (Dynabeads, Invitrogen), followed by incubation for 1 h at 4 °C with gentle agitation. The immunoprecipitated complexes were washed once with sonication buffer, with LiCl buffer (100 mM Tris-HCl, pH 8.0, 500 mM LiCl, 0.1% Nonidet P40 and 0.1% deoxycholic acid), and with Tris-EDTA buffer (TE) pH 8.0 (2 mM EDTA and 50 mM Tris-HCl, pH 8.0), each time for 5 min at 4 °C; this was followed by centrifugation at 4000×*g* for 5 min. Protein-DNA complexes were eluted by incubation with 100 μL of elution buffer (50 mM NaHCO_3_ and 1% SDS) for 15 min at 65 °C. Extracts were incubated for 12–16 h at 65 °C, to reverse crosslinking the proteins digested with 100 μg/ml proteinase K for 2 h at 50 °C, and the DNA was recovered by phenol/chloroform extraction and ethanol precipitation using glycogen (20 μg/mL) as the precipitation vehicle. The anti-H3K27ac antibodies (abcam Ab4729) were used for this protocol.

### 4.5. Determination of mHtt Protein Levels in Brain Tissue

The right cerebral hemisphere was dissected into the different areas of the brain to be analyzed. The tissues were immediately transferred to dry ice and stored at −80 °C for further analysis. The brain samples were homogenized in RIPA with 1% SDS and a protease inhibitor cocktail (1 tablet per 10 mL, Roche complete, EDTA-free, Sigma-Aldrich) using a sonicator with a 10 s pulse at 40% amplitude. Protein concentrations were estimated using the BCA assay (Pierce, Appleton, WI, USA). Cell lysates was denatured under reducing conditions by boiling 30 μg of total protein with 1 M DTT and 4× LDS sample buffer (Invitrogen) at 95 °C for 5 min. SDS-PAGE was performed using Precast 4–15% Bis-Tris gels (Bio-Rad, Hercules, CA, USA). Gels were run in 1× tris-acetate SDS running buffer for 2 h and transferred to a 0.45 μm PVDF membrane using the Turbo Trans-Blot transfer system (Bio-Rad) for 30 min. Membranes were blocked with 5% skim milk in PBS with 0.1% Tween 20 (PBS-T) for 1 h at room temperature and incubated with reconstituted anti-HTT (MAB5374, 1:1000) and anti-b-actin HRP (sc-47778 1:5000) primary antibodies in 5% skim milk overnight at 4 °C.

### 4.6. Soluble and Insoluble Huntingtin Species

The brain samples were homogenized in Soluble lysis buffer (10 mM Tris pH 7.4, 1% Triton-X 100, 150 mM NaCl, 10% glycerol), douncing slowly 30 times in a 1 mL glass douncer. The homogenate was transferred into labeled, ”Insoluble“ tubes and lyse on ice for 1 h. The samples were centrifuged at 4 °C at 15,000×*g* for 20 min. The supernatant was recovered as the ”Soluble“ fraction. The pellet was washed with 500 μL of lysis buffer (×2) and centrifuged at 4 °C at 15,000×*g* for 5 min after each wash. Then, the pellet was resuspended with 150 μL of lysis buffer supplemented with 4% SDS and sonicated each sample for 30 s at room temperature with a probe sonicator at 40% of amplitude (Insoluble fraction). Finally, the insoluble fraction was boiled for 30 min. Protein concentrations were estimated using the BCA assay (Pierce). SDS-PAGE was performed using Precast 4–15 Bis-Tris gels (Bio-Rad). Gels were run in a 1× tris-acetate SDS running buffer for 2 h and transferred to a 0.45 μm PVDF membrane using the Turbo Trans-Blot transfer system (Bio-Rad) for 30 min. Membranes were blocked with 5% skim milk in PBS with 0.1% Tween 20 (PBS-T) for 1 h at room temperature and incubated with reconstituted anti-HTT (MAB5374, 1:1000), and anti-b-actin HRP (sc-47778 1:5000) primary antibodies in 5% skim milk overnight at 4 °C.

### 4.7. Identification of H3K27ac Level in R6/2 Murine Models

Sequencing results were analyzed using a bioinformatic pipeline, similar to the transcriptomic analysis, to identify changes and/or similarities in H3K27ac levels in striatal tissue as follows. First, the quality of the reads in the ChIP-seq data was analyzed via FastQC; next, adapters were removed with Trimmomatic [74]; the Genome index for the GRCm38/mm10 reference genome were generated with a bowtie-build [79] and the alignment of the reads was performed using Bowtie2 [79], sorting by coordinates and transforming into the BAM format through Samtools View and Samtools Sort [80]. We identified enriched regions, and peak annotations were analyzed with MACS3 [81] software using a broad-cutoff of 0.1, the max-gap option of 200, and a *p*-value of 0.05. The acetylation peaks obtained in each sample and those common to both ages under study were analyzed using Bedtools software [82] using the intersect-bedtool. The count of the reads that align against each gene was performed using the Htseq-count [76], setting the strandness option as “no” as contrasted with the infer_experiment script of the RSeqQC package [77] and finally, a differential analysis of count data was performed with the Deseq2 tool [78], considering a *p*-value <0.05.

### 4.8. Quantification of Differentially Acetylated Promoters

Differentially acetylated promoters were identified following the same steps to identify H3K27ac levels. However, after alignment of the reads using Bowtie against the mouse genome version GRCm38/mm10, the count of the reads that align against each promoter was computed using Htseq-count [76], using instead of the general feature file for genes (GTF), a new GTF that contains information related to promoters defined in a region of 1000 bp up and down streams of the TSS (Appendix A). Then, differentially acetylated promoters were detected with the Deseq2 tool [78] over the generated list considering a *p*-value of <0.05.

### 4.9. Determination of CRE Regions and the Genes Associated with These Regions in Mice

CRE-regulated candidate genes were identified using specific distance thresholds from the transcription start site and the occurrence of CREB-binding motif sequences in the genome. To identify CRE-binding regions in the mouse genome, we first downloaded CREB-binding motifs of CRE elements from the JASPAR database [83] scanned in the GRCm38/mm10 version mouse genome for occurrences using FIMO software with default parameters [84]. Subsequently, by creating a script in Python language, we define putative CRE regulated genes by assigning a distance parameter of ±3 Kb from the transcription start site (TSS) (Appendix A).

### 4.10. Generation of Gene Regulatory Networks in the R6/2 Murine Model of HD and Their Respective Control

Considering the expressed genes in both models (R6/2 and their controls at 4 and 6 weeks old), in total, four gene regulatory networks were developed as described. Briefly, we kept outgoing edges of expressed TFs from the union of high confidence general gene regulatory networks (GRNs) deposited in TRRUST [26], RegNetwork [85], and DoRothEA [86], to convert these general networks into context-specific networks. Improbable regulations were filtered by keeping those that arise from TF only if the TF is expressed as previously described [87,88]. Additionally, we have integrated the information related to gene expression and promoter/CRE acetylation status as node/edge attributes. The Cytoscape session that includes these networks can be found in the Appendix A.

### 4.11. Inference of Master Regulators

The Master Regulators (MRs) search in our contextualized networks was performed as follows. For genes defined as differentially expressed, we select their first and second upstream neighbors in both networks similarly [88]. Then, we looked for nodes with the highest edge density according to Davis & Rebay [57]. To do so, an iterative step for deleting nodes with a lower out-degree was performed until the deletion of a node in the subnetwork resulted in an unconnected network, with remaining TFs were categorized as candidates to be MR. To determine if these candidates fulfill the definition of MR [57], we have observed, for candidates’ TFs that regulate the candidate, if its candidate is regulated by other candidates, and if the candidates have a physical interaction annotation among them [89].

## Figures and Tables

**Figure 1 ijms-24-10848-f001:**
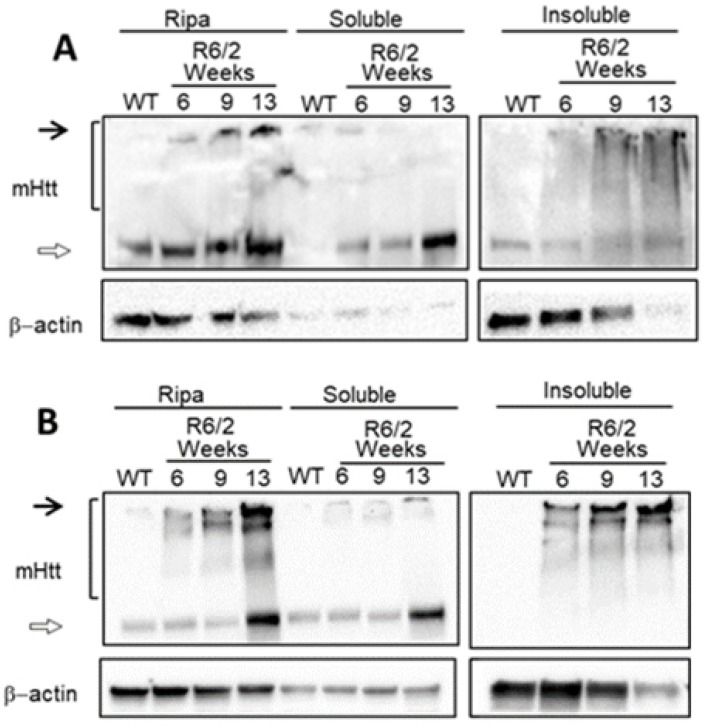
Mutant Huntingtin progressively aggregates for 6 weeks. Total protein extraction was performed, using an RIPA buffer or separating soluble and insoluble fractions from R6/2 mice (6, 9, and 13 weeks old) and WT mice. (**A**) Western blot of striatal brain tissue from WT or R6/2 mice; black arrows indicate high molecular weight species, bracket shows smearing of mutant Huntingtin. (**B**) Western blot from brain cortex tissue from WT or R6/2 mice; black arrows indicate high molecular weight species and bracket shows smearing of mutant.

**Figure 2 ijms-24-10848-f002:**
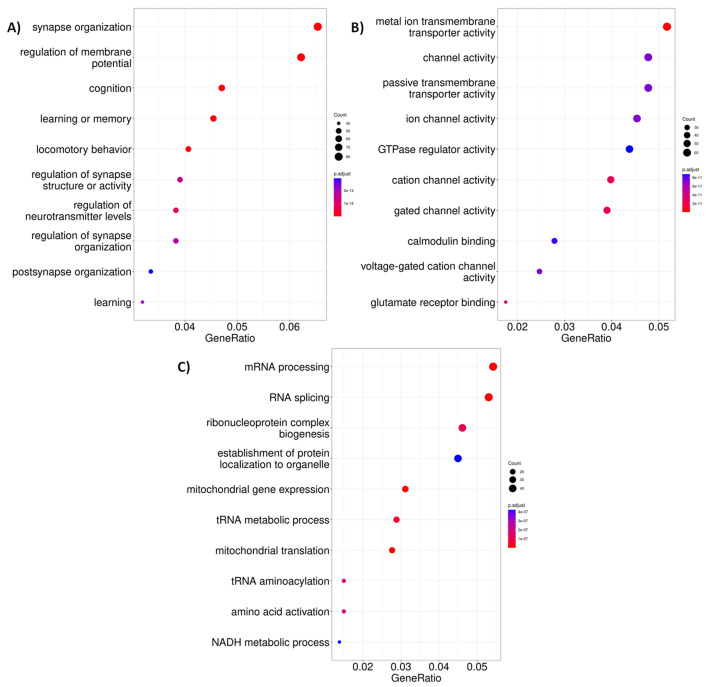
Top 10 enriched GO terms found in differentially expressed genes. (**A**) Biological processes in the 6-week-old R6/2 mice. (**B**) Molecular functions enriched in the 6-week-old R6/2 mice. (**C**) Biological processes for only defined as DE genes in the two-factor analysis.

**Figure 3 ijms-24-10848-f003:**
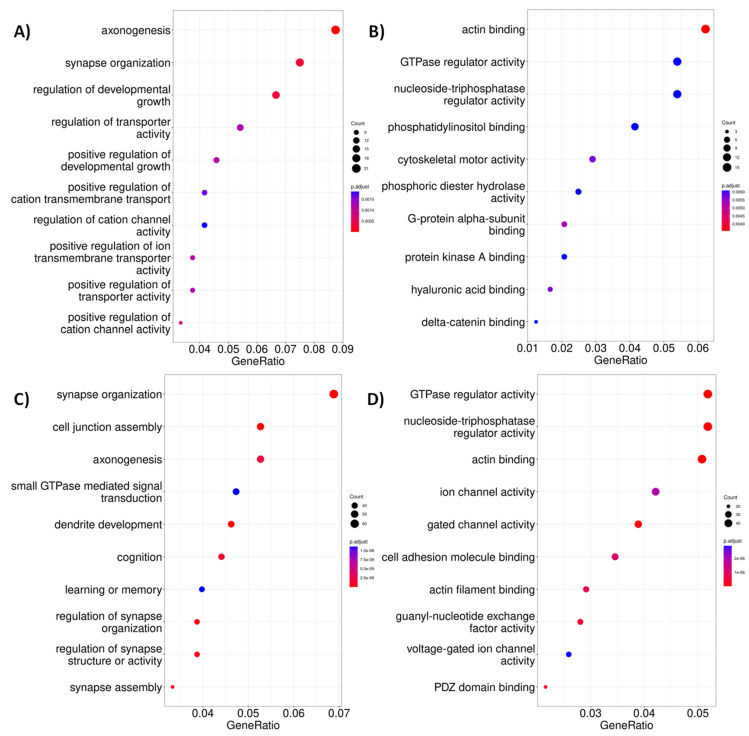
Top 10 enriched GO terms found in genes with an analysisCRE element-with altered levels of H3K27 acetylation. (**A**) Top 10 Enrichment of biological processes and (**B**) Top 10 molecular functions of differentially expressed CRE-region-containing genes observed at four weeks of age. (**C**) Top 10 Enrichment of biological processes and (**D**) Top 10 molecular functions of differentially expressed CRE-region-containing genes at six weeks of age.

**Figure 4 ijms-24-10848-f004:**
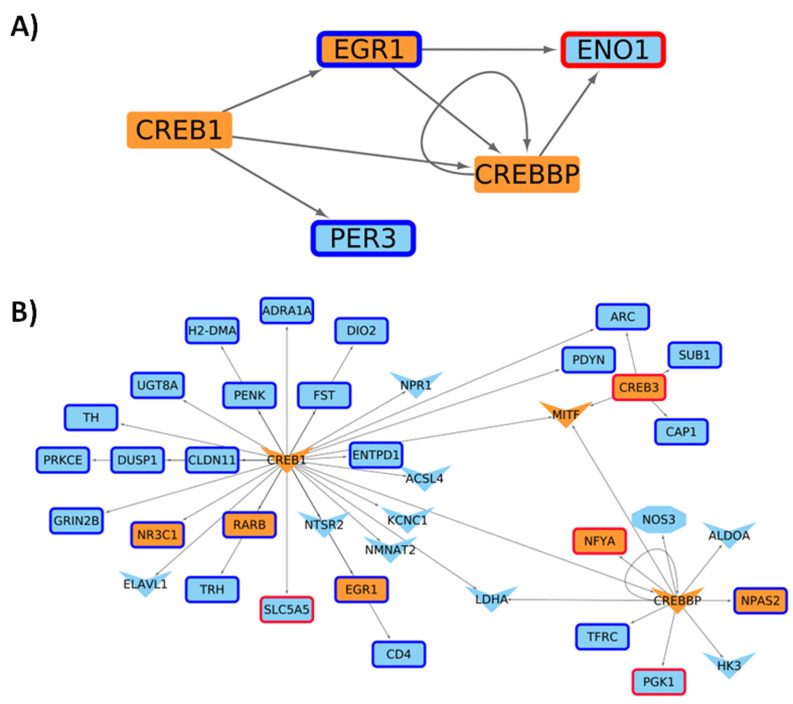
Genes up (red borders) and down (blue border) regulated by CREB/CBP (orange-colored nodes). In (**A**) four week-old mice and (**B**) six week-old mice. V-shaped nodes represent genes with down acetylated promoters. Transcription factor coding genes (orange) presenting low H3K27ac levels in their promoter and CRE regions (V-shape and octagon form) directing genes downregulated in six weeks-old Huntington’s Disease model mice.

**Figure 5 ijms-24-10848-f005:**
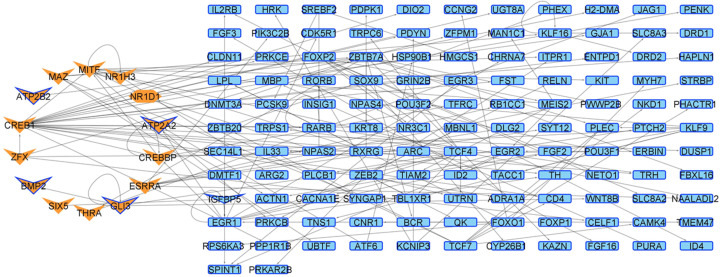
TFs with downacetylated promoters and genes that control. Genes down-regulated (blue border) directed by TF (orange-colored nodes) whose promoters are down-acetylated in 6-weeks mice (V-shape for promoters and octagon form for CRE binding site).

**Figure 6 ijms-24-10848-f006:**
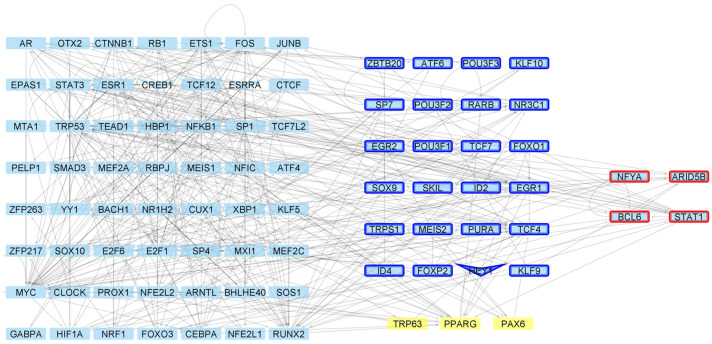
Master Regulators controlling gene expression changes in the HD model at 6 weeks of age. For these MRs in HD condition, there are genes up (**red borders**) and down (**blue border**) regulated, with V-shaped nodes representing genes with down acetylated promoters. Yellow TFs are defined as MRs only under HD conditions.

**Figure 7 ijms-24-10848-f007:**
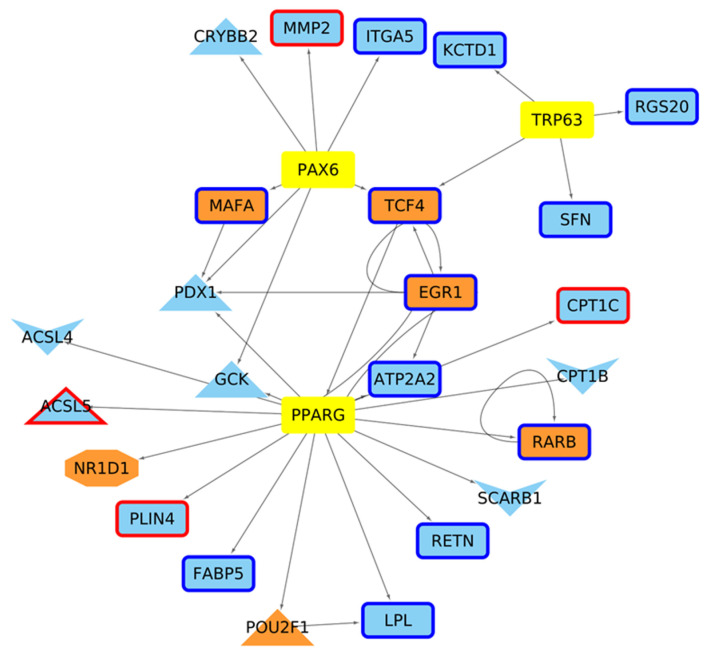
Master Regulators only present in the 6 week mice and genes that direct. For the HD model, there are genes up (**red borders**) and down (**blue border**) regulated, with V-shaped nodes representing genes with down acetylated promoters and the octagon shape represents CRE regulated genes with altered H3K27ac levels. Yellow nodes are the MR only under HD conditions, orange nodes show TFs present, and cyan nodes represent non-coding TFs.

**Figure 8 ijms-24-10848-f008:**
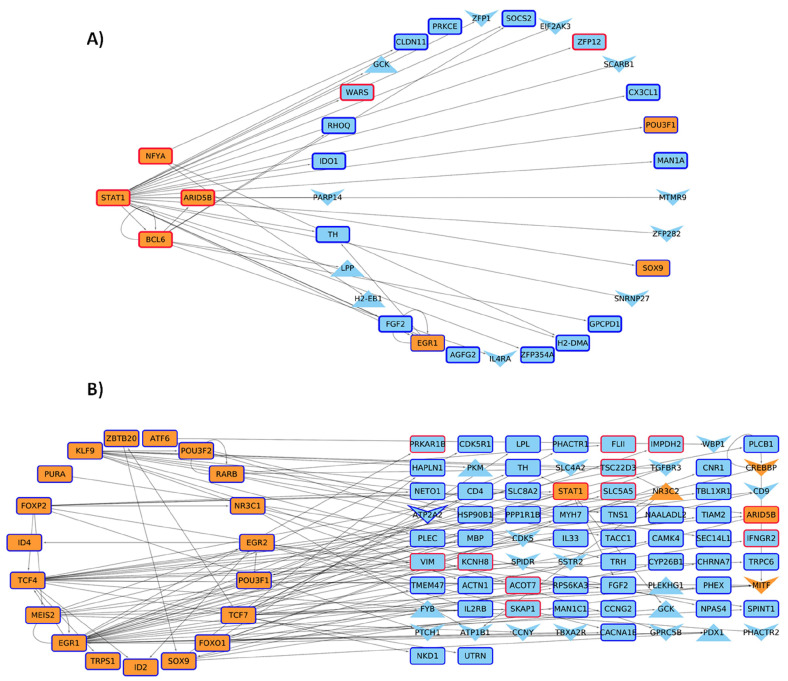
Master Regulators Differentially Expressed. (**A**) Upregulated MR in HD and (**B**) downregulated MR in HD. For theHD model, there are genes up (**red borders**) and down (**blue border**) regulated, with V-shaped and triangle nodes representing genes with down and up acetylated promoters. Orange nodes show TFs present and cyan nodes represent non-TF genes.

**Table 1 ijms-24-10848-t001:** Number of nodes and edges (total and unique) in the GRN derived for mutant and control conditions in 4- and 6-weeks old mice.

Network	Total Nodes (TFs)	Unique Nodes (TFs)	Total Connections	Total Connections
4 weeks Control	5876 (643)	0	16,635	0
4 weeks Mutant	5972 (680)	96 (9)	17,292	657
6 weeks Control	5841 (641)	1 (1)	16,412	22
6 weeks Mutant	5925 (663)	84 (8)	16,990	600

## Data Availability

All sequence data are accessible with accession number BioProject ID: PRJNA873204. https://dataview.ncbi.nlm.nih.gov/object/PRJNA873204?reviewer=q2jn8g48s5gjf7hlo6b7a13h02, accessed on 20 June 2023.

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
