# Peer review of "Transcriptional and Histone Acetylation Changes Associated with CRE Elements Expose Key Factors Governing the Regulatory Circuit in the Early Stage of Huntington’s Disease Models"

_ijms, 2023, doi:10.3390/ijms241310848_

Round 1

Reviewer 1 Report

The authors investigated gene expression and H3K27ac variation in juvenile HD mice, uncovering critical regulators in the disrupted transcription cascade. It identified acetylation and gene expression changes associated with impaired neuronal development, particularly in 6-week-old mice. The manuscript is well-written, and the methods are precise. In addition to studying H3K27ac, it would be beneficial to speculate on the role of other histone modifications, such as H3K9ac and H3K4me3, in Huntington's disease (HD). These modifications may also play significant roles in gene expression regulation and contribute to the pathogenesis of HD. Further investigation into their involvement could provide a more comprehensive understanding of the epigenetic changes underlying the disease. 

Author Response

Dear reviewer,

Thank you for taking the time and effort to review our manuscript titled "Transcriptional and Histone Acetylation Changes Associated with CRE Elements Expose Key Factors Governing the Regulatory Circuit in the Early Stage of Huntington's Disease Models."

We would like to express our gratitude for your valuable feedback. We fully agree on the importance of other chromatin modifications are fundamental to understand the phenomena that are occurring. For this reason, we are currently analyzing the sequencing results of other histone marks, such as H3K4me3 and H3K4me1, which will be part of a subsequent work that will study epigenetics alterations on regulatory elements in HD. According to your comments we included at final segment of Discussion section a paragraph about importance of other histone modifications (line 309).

We want to assure you that we have thoroughly reviewed the entire article to ensure the accurate and coherent implementation of all corrections. Once again, we sincerely appreciate the time and effort you have dedicated to reviewing our work.

Reviewer 2 Report

Arancibia-Opazo et al., describes the changes in epigenetic modification in Huntington’s disease using a mouse model in a pre-symptomatic stage of the disease.

Though they show compelling evidence that the changes in acetylation of chromatin and thereby the transcriptional differences in HD in contrast to control animals, the rationale of CBP binding needs to be substantiated in the current model they have used.

Minor comments

Line 4: CBP acronym is not described.

Line 41: “ER stress and abnormal protein-protein interactions [22]” This referred article does not describe protein-protein interaction or ER stress.

Line 43: altered gene expression pattern into the nucleous? What does this mean?

Line 63: Pre-symptomatic

Line 74 and 80: Striated tissue does it mean striatal?

Line 94: 100 base pair-end sequencing?

Line 111: Why tests?

Line 123: 30 seconds On and 30 seconds Off?

Line 197: 1000 bp up or downstream. The unit base pair need to be mentioned.

Line 239: The data is shown for 6, 9, and 13 weeks, not 4 weeks.

Line 267: Has this been shown in R6/2 model? A reference to this would be ideal if not from the current study.

Line 383: What does this mean? could be rewritten.

Line 294: Synapsis? Synapses.

Line 321-323: Not clear what this means. could be rewritten.

Figure 8 figure legend: “V-shaped an triangle nodes representing genes with down and up acetylated promoters” Not clear what this means. “cyan nodes represent non-coding TFs” What do non-coding TFs mean? non-TF genes?

Figure A1: Text in the figure and the legend needs to mention basepair bp instead Pb.

Figure A7: Could you please comment on why the 6-week acetylation does not separate in the PCA plot, though it is significantly changed?

Figure A12: No diamond shape or triangle in the figure.

Author Response

Dear reviewer,

Thank you for taking the time and effort to review our manuscript titled "Transcriptional and Histone Acetylation Changes Associated with CRE Elements Expose Key Factors Governing the Regulatory Circuit in the Early Stage of Huntington's Disease Models."

We would like to express our gratitude for your valuable feedback. We accepted and included all your minor comments, as you can check in the attached file. Also, while it is true that CBP has a well-established role in Huntington's Disease and its involvement in the acetylation of H3K27ac, we understand the need to provide supporting evidence within the context of the model. To address this concern, we have included a brief explanation in the introduction that highlights the connection and relevance between the model and CBP. We added two new cites and a sentence in Introduction to clarify that CBP aggregation is also present in R6/2 model (line58) and how to reduce its aggregation improve the animal condition in R6/2 model.

We want to assure you that we have thoroughly reviewed the entire article to ensure the accurate and coherent implementation of all corrections. Once again, we sincerely appreciate the time and effort you have dedicated to reviewing our work.
